# Structural Relationship among Physical Self-Efficacy, Psychological Well-Being, and Organizational Citizenship Behavior among Hotel Employees: Moderating Effects of Leisure-Time Physical Activity

**DOI:** 10.3390/ijerph17238856

**Published:** 2020-11-28

**Authors:** Ji-hoon Kang, Yun-ho Ji, Woo-yeul Baek, Kevin K. Byon

**Affiliations:** 1Department of Physical Education, Chuncheon National University of Education, Gongji-ro 126, Korea; 76015020@hanmail.net; 2Department of Tourism Administration, Kangwon National University, Gangwondaehak-gil 1 KR, Korea; 3School of Sport Science, Kyonggi University, Gwanggyosan-ro 154-42, Korea; wyb71@kyonggi.ac.kr; 4Department of Kinesiology, Indiana University, Bloomington, IN 47405, USA; kbyon@indiana.edu

**Keywords:** physical self-efficacy, psychological well-being, organizational citizenship behavior, leisure-time physical activity, hotel employees

## Abstract

Critics argue that service firms should pay more attention to human resource management’s psychological and voluntary aspects to contribute to overall organizational development. The purpose of this study was to investigate the effects of physical self-efficacy on the psychological well-being and organizational citizenship behavior among hotel employees and the moderating effects of leisure-time physical activity on the relationships between the previously mentioned variables. To achieve the research purpose, 346 hotel employees working at the room, food, beverage, and kitchen departments of 10 hotels located in Seoul, South Korea, participated in the study. The researchers visited their department meetings and provided a brief description of the present study and informed consent forms to participate in the study. After obtaining written informed consent forms, the researchers distributed the surveys and asked participants to complete them. Several statistical analyses, including descriptive statistics, confirmatory factor analysis (CFA) for examining the hypothesized model’s psychometric properties, and structural equation modeling (SEM) for testing the hypotheses were conducted using SPSS Ver. 23.0 and AMOS 23.0. Results revealed that perceived physical ability and self-presentation confidence, and psychological well-being positively affected organizational citizenship behavior. Perceived physical ability also had a positive effect on psychological well-being. Lastly, leisure-time physical activity had a partial moderating role in the relationships between the variables mentioned above. This study suggests that promoting employees’ participation in leisure-time physical activity is needed to improve service workers’ organizational citizenship behavior via physical self-efficacy and psychological well-being enhancement.

## 1. Introduction

Due to the uncertainty of market environments, service firms are more dependent on their employees’ voluntary participation, which refers to a form of employees’ organizational citizenship behavior that voluntarily helps other coworkers beyond their defined work [1], and commitment than ever before. The importance of the capability maximization of human resources to effectively cope with the environmental changes is increasing. Service firms, which actively interact with customers, have been focusing their organizational capabilities on members’ organizational immersion and enhancing their customer-oriented attitudes [2]. The environmental adaptability of a service company depends on its employees, and their voluntary efforts are critical to achieving sustainable competitive advantages. Thus, innovating a service firm to respond to rapidly changing market environments requires considering their employees’ competencies. Self-efficacy enhancement may be one of the most critical determinants of creating sustainable competitive advantages [3].

According to Kim and Byon [4], self-efficacy refers to individuals’ belief that they can successfully perform a given specific task. In the service industry, workers cannot correctly predict what conflicts and risks are lurking with customers and are consequently exposed to high tension levels at work. Service workers often experience psychological exhaustion due to excessive work stress, lack of psychological well-being, and low job satisfaction [5]. Since service products are mainly produced and delivered by service workers, their belief that a given task will lead to success can be an essential psychological variable for quality services [6]. As a result, service workers’ self-efficacy and psychological well-being seem crucial in shaping voluntary work attitudes and behaviors.

Moreover, recent studies on organizational performance management have explored the importance of organizational citizenship behavior that increases an entire organization’s productivity and efficiency [7,8]. According to Scholz et al. [9], service firms need to encourage their employees to have an energetic and voluntary attitude toward a given task to improve organizational performance. Spector and Fox [8] also stressed that when the employees perceive a high self-efficacy toward the task performance, they are likely to reduce counterproductive work behavior and increased organizational citizenship behavior. Additionally, Paul et al. [7] argued that employees’ subjective well-being was a significant predictor of organizational citizenship behavior among workers in the Indian manufacturing industry. Thus, service firms should pay more attention to human resource management’s psychological and voluntary aspects to contribute to overall organizational development.

Meanwhile, leisure-time physical activity, which refers to all actions associated with physical activity in which people participate in freely disposing time, is associated with self-efficacy and psychological well-being. For instance, Molina-Garcia et al. [10] reported that university students with higher leisure-time physical activities experienced higher vitality and self-esteem than counterparts. Accordingly, the researchers concluded that high leisure-time physical activity is closely linked with psychological well-being. Liu and Dai [11] also found that Chinese university students participating in leisure-time physical activity had higher physical efficacy than those who did not. Additionally, Kekäläinen et al. [12] reported that walking was positively related to psychological and social well-being. They also found that endurance training was associated with subjective health and well-being. The results suggest that leisure activities were associated with psychological well-being. Collectively, the levels of leisure-time physical activity are positively correlated with self-efficacy and well-being. However, despite voluminous previous research on self-efficacy, most of them have focused on the relationships among self-efficacy, psychological, and/or job performance aspects, neglecting moderating effects of employees’ characteristics, such as physical activity levels during their free time. If considering such employees’ characteristics, we might fill the research gaps of relevant extant studies [3,9] in explaining the service workers’ psychological and job performance enhancement.

As such, the current study aimed to investigate the effects of physical self-efficacy on the psychological well-being and organizational citizenship behavior among hotel employees and the moderating effects of leisure-time physical activity on the relationships between physical self-efficacy, psychological well-being, and organizational citizenship behavior. It is important to note that we wanted to differentiate the present study from the extant research by exploring the moderating roles of leisure-time physical activity in the relationships between the aforementioned variables. In fact, hotel employees are faced with a heavy workload and low wages and are commonly required to do excessive emotional labor, which affects turnover, while enduring consumers’ mistreatment [13,14]. In this study, we attempted to shed light on how hotel employees’ perceived physical self-efficacy affects psychological well-being and organizational citizenship behavior. We also examined if the effects of perceived physical self-efficacy on physical well-being and organizational citizenship behavior differ based on employees’ leisure-time physical activity levels. The current study’s findings are expected to provide hotel organizations with effective measures to manage human resources by considering leisure-time physical activity programs and the psychological well-being of employees.

## 2. Literature Review and Hypotheses Development

### 2.1. Physical Self-Efficacy, Psychological Well-Being, and Organizational Citizenship Behavior

From social cognitive theory, which assumes that learning occurs in a social context through dynamic and interactive interactions of people, environments, and actions, self-efficacy is the belief of an individual in his or her ability to complete a certain task [15] successfully. Similarly, physical self-efficacy, which is a form of self-efficacy, refers to an individual’s perceived level of competence related to physical tasks [16]. Interestingly, physical self-efficacy is also used as an interventional psychological mechanism to improve exercise participation and immersion. According to McAuley and Blissmer [17], physical self-efficacy consists of perceived physical ability, which refers to the ability to perform physical skills and tasks, and physical self-presentation confidence individual’s belief in demonstrating physical skills to be evaluated by others. Interestingly, perceived physical ability and physical self-presentation confidence increase individuals’ subjective efficacy, such as speed, intensity, and reaction time. Physical self-efficacy affects what activity a person chooses, how much effort the person should make, and how much persistence the person should have in the face of difficulties, which determines the present and future behaviors [18].

Well-being is harmonious satisfaction between one’s desires and goals. Psychological well-being refers to a sense of subjective satisfaction, satisfaction with life experiences, one’s role in the world of work, a sense of accomplishment, and a greater sense of belonging [9]. The subject of well-being has been the center of many scholars’ interests over the past decades. It includes various components and aspects, such as psychological, physical, economic, and social well-being [3]. Especially, Ryff [19] introduced six sub-dimensions of psychological well-being, including self-acceptance, positive relationship with others, autonomy, environmental mastery, life purpose, and personal growth. The research found that these subcomponents of psychological well-being were shown consistently across different environments. Additionally, recent studies in the organizational behavior context have recognized psychological well-being as the antecedent of employee job performance and work efficiency [3,20].

Organizational citizenship behavior can be defined as the discretionary behavior of individuals that effectively promotes an organization [1]. Since the term of organizational citizenship behavior (OCB) was first coined by Organ and his colleagues [21], scholarly concern in OCB has dramatically increased [22]. Interestingly, Niehoff and Moorman [23] subdivided organizational citizenship behavior into five distinct categories: altruism, courtesy, sportsmanship, conscientiousness, and civic virtue. Williams and Anderson [24] also distinguished organizational citizenship behavior into organizational citizenship behavior-individual (OCB-I), which is given immediate benefits to an individual, and then indirectly to an organization, and organizational citizenship behavior-organization (OCB-O), which benefits the organization as a whole. OCB-I includes altruistic and respectful behavior that benefits individuals in the organization, while OCB-O refers to legitimate behavior that directly helps the organization. Ultimately, OCB-O can be understood as an essential measure that positively impacts maintaining and developing an organization. The present study focused on OCB-O, which is directly related to organizational effectiveness and performance. As such, thisstudy’s primary purpose was to identify the effects of physical self-efficacy on the psychological well-being and organizational citizenship behavior among hotel employees.

Bandura [18] emphasized self-efficacy as a belief in one’s ability to organize and execute the course of action necessary to achieve goals. Self-efficacy is an important predictor of work-related well-being, such as job satisfaction and psychological happiness [3]. Physical self-efficacy, which consists of perceived physical ability and physical self-presentation confidence, has been reported to reduce negative emotional states such as anxiety and depression and improve self-concept, self-esteem, and cognitive ability [17]. Given this, Darvishmotevali and Ali [3] found that hotel employees’ psychological capital, which includes physical self-efficacy, optimism, resilience, and hope, has a buffering role in the relationship between well-being and job performance. As such, they concluded that self-efficacy is positively correlated with self-confidence and respect, positive emotions, well-being, mental and physical health, and adaptation to difficult situations. Additionally, Niu [25] reported the positive impacts of foodservice employees’ creative self-efficacy on job crafts and job satisfaction. The finding indicates that the more employees have beliefs in their abilities to achieve creative job performance, the more they positively change their relationships with others at work and their perceptions of their jobs and the higher job satisfaction. Based on the findings mentioned above from prior research, the following hypotheses are proposed:

**Hypothesis 1.** 
*Perceived physical ability has a positive effect on psychological well-being among hotel employees.*


**Hypothesis 2.** 
*Physical self-presentation confidence has a positive effect on psychological well-being among hotel employees.*


Physical self-efficacy affects athletic performance and provides the basis for motivation, well-being, and individual performance. It is closely associated with the thinking patterns and emotional reactions of an individual [16]. Early self-efficacy levels affect the willingness and ability to perform tasks independently within a group environment [26]. Moreover, Chen and Kao [27] reported that physical self-efficacy positively affected Taiwanese police officers’ organizational citizenship behavior. Spector and Fox [8] also argued that when the members of an organization feel full of energy, both psychologically and physically, they can achieve high levels of task performance and control negative emotions to reduce counterproductive work behavior and promote organizational citizenship behavior. Thus, we suggest the following hypotheses:

**Hypothesis 3.** 
*Perceived physical ability has a positive effect on organizational citizenship behavior among hotel employees.*


**Hypothesis 4.** 
*Physical self-presentation confidence has a positive effect on organizational citizenship behavior among hotel employees.*


Psychological well-being is closely related to social relationships, emotional feelings, or cognitive judgments about the surrounding environment. Since service ”workers’ positive emotions, such as pleasant mood and satisfaction, can promote social identity, community consciousness, and work effectiveness, their emotions are closely linked to organizational performance [28]. Paul et al. [7] found that employees’ subjective well-being was a significant predictor of organizational citizenship behavior in the Indian manufacturing industry. Moreover, Van Katwyk et al. [29] reported that university employees who experienced well-being in the organization took pro-social and organizational citizenship behaviors and showed high voluntary level participation. Thus, we propose the following hypothesis:

**Hypothesis 5.** 
*Psychological well-being has a positive effect on organizational citizenship behavior among hotel employees.*


### 2.2. Moderating Effects of Leisure-Time Physical Activity

Participating in sport and physical activities improve participants’ mental as well as physical health. In particular, leisure-time physical activity provides participants with pleasure through competition and social interactions, eliminating stress from their work, and monotonous daily lives. Liu and Dai [11] reported that Chinese university students participating in leisure-time physical activity showed higher physical efficacy than those who did not. As such, the researchers concluded that the level of leisure-time physical activity is significantly correlated with self-efficacy. Molina-Garcia et al. [10] also found that university students with higher leisure-time physical activities experienced higher vitality and self-esteem than those with low activities, indicating that psychological well-being is determined by high leisure-time physical activity for college students. Based on the findings mentioned earlier from prior research, the following hypothesis is suggested (see Figure 1 for a graphical representation of the research model):

**Hypothesis 6.** 
*The effects of perceived physical ability and physical self-presentation confidence on psychological well-being and organizational citizenship behavior will be stronger for hotel employees who are engaged with active leisure-time physical activities compared to those who are not.*


**Figure 1 ijerph-17-08856-f001:**
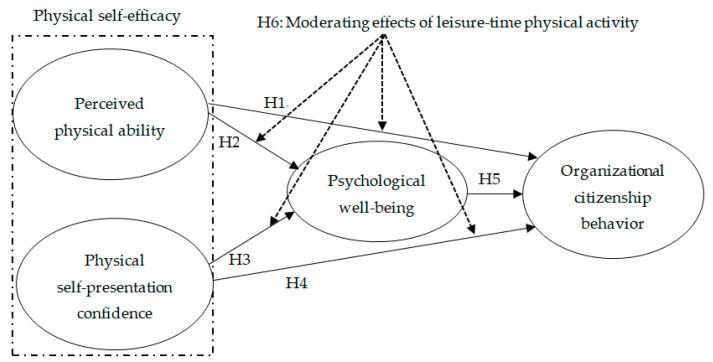
Research model.

## 3. Methods

### 3.1. Participants

Using a convenience sampling method, the researchers contacted the head managers of 10 hotels located in Seoul, South Korea, via cellular phones. They acquired permission to attend the department meetings, including room, food and beverages, and kitchen departments. At the end of each session, the researchers provided a brief description of the present study and informed consent forms to participate in the study. After obtaining written informed consent forms, the researchers distributed the surveys and asked participants to complete them. The researchers collected the surveys right after the participants completed them.

Of 362 surveys hotel employees sampled, 346 (95.6%) returned fully completed surveys. Female participants accounted for 37.9%, and the majority of participants were in their 40 s and older (77.6%) and affiliated with the food and drink cooking department (62.7%). Most participants participated in leisure-time physical activity for 4 h and longer (77.5%). The summary of the participants’ demographics is presented in Table 1.

Additionally, the study subjects who participated in leisure-time physical activity “almost every day”, “2 or 3 times a week”, and “once a week” were considered active exercise participants (*n* = 162) while those who engaged in physical activity “less than one time a month” and “never” were considered non-active exercise participants (*n* = 184) [30].

### 3.2. Instruments

The survey questionnaire consisted of 24 items designed to measure physical self-efficacy, psychological well-being, organizational citizenship behavior, and leisure-time physical activity (see Table 2). The survey items on participants’ demographic information include gender, age, departmental affiliation, work position, and leisure-time physical activity participation levels. All items except for demographics were adapted from the previously validated scales described below, in which a 5-point Likert scale was used, ranging from 1 (strongly disagree) to 5 (strongly agree).

Physical self-efficacy consisted of perceived physical ability (five items), and physical self-presentation (five items) was assessed with ten items taken from Ryckman et al. [16]. The exemplary item of perceived physical ability was, “My physique is rather strong”. One measure of physical self-presentation was, “I do not have any physical defects that bother me”. Psychological well-being, which refers to the satisfaction of objective conditions and the area of personal living, was measured with five items taken from Ryff [31]. The exemplary item of psychological well-being was, “I have a positive attitude toward myself”. Lastly, organizational citizenship behavior, which refers to a legitimate action that directly benefits the organization without practical reward, was measured with four items adapted from Niehoff and Moorman [23]. One measure of organizational citizenship behavior was “I attend and participate in meetings regarding the organization”.

Regarding the translation process for the scale developed in the present study, we followed Brislin’s [32] guideline. First, a native Korean speaker fluent in English translated the original English version scales into Korean. After that, a scholar who was also fluent in Korean and English back-translated the translated version into English. There were no discrepancies between the two versions. The items translated from English into Korean through back-translation process were reviewed by four professors whose expertise lies in hospitality management, sport management, and measurement reviewed the items’ relevance, representativeness, and clarity.

### 3.3. Data Analysis

We used Anderson and Gerbing’s [33] two-step modeling approach, in which confirmatory factor analysis (CFA) was first performed to estimate the measurement model, followed by a structural equation modeling (SEM) to test the hypothesized model. For estimating the measurement model, we used multiple model fit indexes (i.e., GFI > 0.90; AGFI > 0.90; CFI > 0.90; NFI > 0.90, RMSEA < 0.08, and RMR < 0.08; [34,35]. Factor loading (>0.70, [33]) and average variance extracted (AVE > 0.50, [36]) were used to examine convergent validity and discriminant validity was assessed via comparing squared correlation with AVE value [36]. Reliability was assessed with the calculation of composite reliability (>0.70; [36]).

To test hypothesis 6 (i.e., moderating effect of leisure-time physical activity), an invariance test was performed. First, metric invariance was tested to examine if items representing the latent variables operate similarly across the two groups. Upon establishment, structural invariance was carried out to test the moderating effect of leisure-time physical activity. All statistical analyses in the present study were performed using SPSS 23.0 (IBM Corp, Armonk, NY, USA) and AMOS 23.0 (IBM Corp, Chicago, IL, USA). 

## 4. Results

### 4.1. Measurement Model

Based on the initial confirmatory factor analysis (CFA), the factor loading of one item measuring perceived physical ability failed to achieve the required level of 0.70 and removed from the proposed model. As a result, a total of 23 question items were used for subsequent CFA, resulting in an acceptable model fit (χ^2^ = 194.01, *df* = 129, GFI = 0.93, AGFI = 0.91, NFI = 0.94, CFI = 0.97, RMR = 0.05, and RMSEA = 0.03; Hair et al.). All factor loadings were statistically significant (*p* < 0.05) and ranged from 0.70 to 0.86, and average variance extracted (AVE) values ranged from 0.51 to 0.54, indicating good convergent validity. Discriminant validity was supported, as all AVE scores were higher than the squared correlations of all pairs of variables [33]. Composite reliability values were all over 0.70, indicating good reliability of the constructs (see Table 2). All factors of means, standard deviation, and correlations are also shown in Table 3.

### 4.2. Structural Equation Modeling

As shown in Table 4, the structural model indicated an adequate model fit to the data (χ^2^ = 194.01, *df* = 129, CFI = 0.97, TLI = 0.97, RMR = 0.04, and RMSEA = 0.03). The result of SEM showed that perceived physical ability positively predicted psychological well-being (*β* = 0.73, *p* < 0.001) and organizational citizenship behavior (*β* = 0.73, *p* < 0.001). Additionally, physical self-presentation confidence had a significant effect on organizational citizenship behavior (*β* = 0.31, *p* < 0.001), but not on psychological well-being (*β* = 0.03, *p* > 0.05). Lastly, psychological well-being was positively related to organizational citizenship behavior (*β* = 0.73, *p* < 0.001) (see Figure 2).

### 4.3. Multi-Group Analysis

To examine if the moderating effect of leisure-time physical activities on the relationship between physical self-efficacy and psychological well-being as well as physical self-efficacy and organizational citizenship behavior, a multi-group analysis was conducted. The pooled data were split into two groups: (a) active exercise participants (*n* = 162) and (b) non-active exercise participants (*n* = 184). A multi-group analysis revealed that the difference in χ^2^ between the unconstrained model (χ^2^(328) = 490.42) and the constrained model (χ^2^(354) = 562.87) was found to be statistically significant (Δχ^2^(26) = 72.45, *p* < 0.001), indicating the leisure-time physical activities played a moderating role.

To discover where the statistical differences exist between the two groups, each path coefficient was compared between two groups via a multi-group analysis. The path coefficient for the relationship between perceived physical ability and psychological well-being was found to be statistically different across the groups (Δχ^2^(1) = 5.60, *p* < 0.05). The relationship strength was found to be stronger in the active group (*β* = 0.46, *p* < 0.001) than in the non-active group (*β* = 0.12, *p* < 0.05). The result for the relationship between physical self-presentation confidence and organizational citizenship behavior was also found to be significant across the two groups (Δχ^2^(1) = 9.66, *p* < 0.001). The relationship strength was stronger in the active group (*β* = 0.44, *p* < 0.001) than in the non-active group (*β* = 0.09, *p* > 0.05).

However, the magnitude of the relationship between physical self-presentation confidence and psychological well-being was not found to be significant between the groups (Δχ^2^(1) = 1.56, *p* > 0.05). Nonetheless, as predicted, the relationship strength was found to be stronger in the active group (*β* = 0.40, *p* < 0.001) than in the non-active group (*β* = 0.23, *p* < 0.05). Lastly, there was no significant difference in the relationship between perceived physical ability and organizational citizenship behavior (Δχ^2^(1) = 2.16, *p* > 0.05). The results of the multi-group analysis indicate that hypothesis 6 was partially supported (see Table 5).

### 4.4. Common Method Variance

We tested if the results were subjected to common method variance (CMV), which is defined as “variance that is attributable to the measurement method rather than to the constructs the measures represent” [37] due to two reasons: (a) data were collected from the same source (i.e., head managers of 10 hotels) even though heterogeneity of the data was ensured, in which the data were collected from 10 hotels and (b) independent, mediator, and dependent variables were measured simultaneously. To this end, we applied Harman’s single factor method [38]. We used exploratory factor analysis (EFA) using principal component analysis (PCA) with an unrotated factor solution. The results showed that the first factor explained a total of 28% of the variance. According to Andersson and Bateman [38], CMV would be an issue if the total variance explained by the first factor is over 50%. Therefore, we concluded that CMV is not an issue with the data set used in this study.

## 5. Discussion

The current study aimed to investigate the impacts of employees’ physical self-efficacy on the hotel industry’s psychological well-being and organizational citizenship behavior. Results revealed that perceived physical ability and self-presentation confidence, and psychological well-being positively affected organizational citizenship behavior. However, only perceived physical ability had a positive effect on psychological well-being. In addition, leisure-time physical activity had a partial moderating role in the relationships between these variables. These findings lead to several implications and are expected to provide hotel organizations with effective measures to manage human resources by considering leisure-time physical activity programs as well as the psychological well-being of employees.

The present study revealed that perceived physical self-efficacy well predicted the well-being of hotel workers. This study’s finding is consistent with the previous studies of Darvishmotevali and Ali [3] and Niu [25], who reported that self-efficacy was a crucial predictor of psychological well-being. However, physical self-presentation confidence did not have any effects on psychological well-being among hotel employees. It is possible that since the labor of most hospitality industries, such as hotels, is made up of tough emotional labor. Service workers strive to provide quality services through emotional control and interactions at the point of contact with customers. They are not likely to perceive physical self-presentation confidence as a prerequisite for determining their well-being [39,40]. Thus, it might be necessary for hospitality firms to provide their workers with a working environment that can enhance perceived physical ability, which leads to an increased sense of psychological well-being while minimizing an organizational atmosphere that can adversely affect the employees.

Another implication is that both perceived physical ability and physical self-presentation confidence had significant positive effects on hotel employees’ organizational citizenship behavior. The present study’s findings are in line with previous research [41,42], indicating that self-efficacy acts as a predictor of essential mechanisms that induce employees’ organizational citizenship behavior. For example, Jung and Yoon [41] reported that psychological capital, operationalized as self-efficacy, hope, optimism, and resiliency, positively impacted Korean hotel employees’ organizational citizenship behavior. They concluded that positive psychological capital is a critical performance factor that improves hotel employees’ attitudes and the organization’s effectiveness. Kao [42] also reported that service workers’ self-efficacy positively affects individual service-oriented organizational citizenship behaviors. Thus, the present study’s findings confirmed that perceived physical ability and self-presentation confidence, which are the sub-components of self-efficacy in the present study, are crucial antecedents of organizational citizenship behavior among hotel employees.

The present study also revealed that service ’workers’ psychological well-being is associated with organizational citizenship behavior. The present study’s findings are consistent with the previous studies of Paul et al. [7] and Ariza-Montes et al. [20]. They reported that employees’ well-being was a significant predictor of organizational citizenship behavior. Since individuals tend to experience psychological well-being in healthy social relationships, an organization’s employees can improve psychological well-being through positive interactions with other workers [31]. Organizational stimulus and employee interactions complement each other because these variables can help individuals improve happiness, life satisfaction, and morale [28]. Therefore, hospitality firms should manage any discord or negative sentiment between employees and organizations or between members that may jeopardize ’employees’ well-being as soon as possible.

Meanwhile, in the regular leisure-time physical activity group, perceived physical ability on psychological well-being and physical self-presentation confidence on organizational citizenship behavior were higher than the non-regular leisure-time physical activity group, partially supporting hypothesis 6 of the current study. This finding was consistent with the results reported in Molina-Garcia et al. [10] and Liu and Dai’s [11] studies. Hospitality services inherently require frequent and intimate interactions with customers, and employees play an essential role in achieving customer satisfaction and loyalty by creating a memorable experience for customers [43]. Improving hospitality ’employees’ wellness is closely linked with their emotional attachment to a firm, customer service performance, and, ultimately, the company’s profitability and productivity [44]. The findings of this study indicated that hotel employees who participated in regular leisure-time physical activities evaluated their self-efficacy higher. In turn, its impacts on psychological well-being and organizational citizenship behavior were stronger than those who did not. Thus, hotels need to recognize the importance of delivering quality services to employees, particularly by instituting long-term wellness programs, which would help the employees to improve their psychological well-being. Ultimately, this internal service management would lead to overall organizational success.

## 6. Conclusions

Hotel employees provide customers with various services such as reception, room information, luggage transportation, room reservation, room arrangement, laundry supply, and food provision. Hotel employees are faced with many challenges while performing their jobs. Facing a heavy workload, frequent changes in circumstances, lack of performance feedback, and low wages, they get easily irritated and exhausted affecting their behavior. They may lead to turnover [14]. Moreover, hotel workers have a high intensity of emotional labor in daily-based customer service interactions regardless of their inner psychological state to comply with organizationally mandated emotional display rules [13]. It might be challenging to expect employees to help coworkers and superiors under such a stressful environment and do devotional actions for hotel organizations. The findings of this study verified that self-efficacy is an essential factor that could enhance hotel ’employees’ psychological well-being and adopt voluntary attitudes at work. Leisure activity strengthens the anteceding role of self-efficacy for the employees’ psychological wellness and devotional behaviors toward the organization. Accordingly, hotel organizations should seek diverse support programs such as counseling and physical activity programs from a managerial standpoint. More specifically, a group counseling program adopting positive psychology can create individuals’ positive emotions when employees are faced with a work-site crisis caused by psychological and non-psychological changes, and they can learn to respond appropriately and flexibly to problems.

Additionally, hotel organizations can provide their employees with an in-house fitness center and a long-term wellness program, which allows them to adopt an active, healthy lifestyle (i.e., the balance of physical, emotional, and social health). Thus, providing such programs to employees should be considered a critical success factor for hotel organizations in the highly competitive hospitality industry. In conclusion, it may be of great importance to understand hotel ’employees’ psychological state in the work environment, provide a variety of psychological counseling and physical activity programs, and support the positive organizational culture and atmosphere.

Findings of this study also suggest that physical self-efficacy is a crucial antecedent of well-being and organizational citizenship, which leads to high employee productivity in the hotel industry. This indicates that hotel managers should pay attention to this personal trait in the recruitment process. Thus, hotel organizations are required to establish a reliable and valid physical self-efficacy test and apply it to employee recruitment and counseling program adjustment.

Despite the contribution of the present study to the organizational psychology context, limitations should be acknowledged. First, generalizing the results to employees working in other hospitality industries needs some caution as the data were collected only from one hospitality stakeholder (i.e., hotel employees). Future studies should apply the tested model to other hospitality contexts. Second, the present study found physical self-efficacy and psychological well-being as the predictors of organizational citizenship behavior, promoting organizational efficiency. It can be suggested that if the present study is replicated, other psychological factors, such as ego-resiliency and positive emotion, should be considered. Third, since the perception toward work environment and workload for hotel employees may be different, it would be interesting to see if the effects of perceived self-efficacy on psychological well-being and organizational citizenship behaviors differ by hotel employees’ work responsibilities. Lastly, because the data analyzed in the current study were collected in the pre-COVID era (i.e., summer of 2018), interpretation of the results requires caution as hotel employees’ perception of’ the leisure-time activity may not be the same as that is during or after the COVID era.

## Figures and Tables

**Figure 2 ijerph-17-08856-f002:**
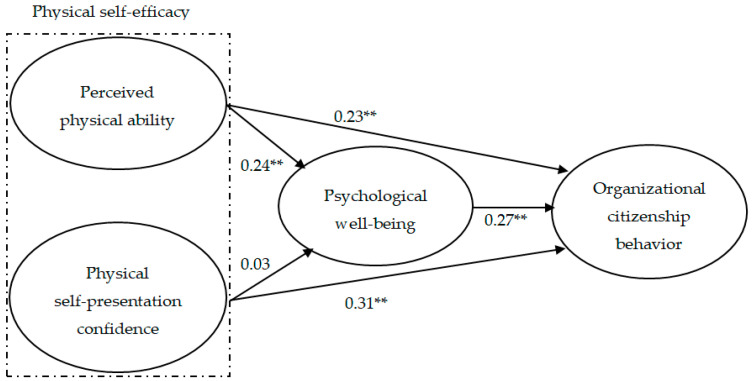
Path coefficients between physical self-efficacy, psychological well-being, and organizational citizenship behavior. ** *p* < 0.01

**Table 1 ijerph-17-08856-t001:** Demographics of participants.

Factors	Categories	Frequency
Gender	Male	131 (37.9%)
	Female	215 (62.1%)
Age	18–30	81 (23.4%)
	30–39	124 (35.8%)
	40–49	84 (24.3%)
	50+	57 (16.5%)
Departmental affiliation	Room	63 (18.2%)
	Food and Beverages	125 (36.1%)
	Kitchen	92 (26.6%)
	Others	66 (19.1%)
Job position	Rank and File	201 (58.1%)
	Supervisor	75 (21.7%)
	Assistant Manager	41 (11.8%)
	Manager	21 (6.1%)
	Director or above	8 (2.3%)
Levels of leisure-time physical activity	Regular participant	162 (46.8%)
	Non-regular participant	184 (53.2%)

**Table 2 ijerph-17-08856-t002:** Factor loadings (λ), composite reliability (CR), and average variance extracted (AVE).

Variables	Items	λ	CR	AVE
Perceived physical ability	I have excellent reflexes.	0.75	0.82	0.53
	My physique is relatively strong.	0.83		
	I have a strong grip.	0.77		
	I have little pride in my ability in sports.	0.75		
Physical self-presentation confidence	I hold up well under stress.	0.86	0.85	0.54
	I do not have any physical defects that bother me.	0.70		
	I am not concerned with the impression my physique makes on others.	0.82		
	Athletic people usually do not receive more attention than me.	0.82		
	I am not hesitant about disagreeing with people bigger than me.	0.75		
Psychological well-being	I am self-determining and independent.	0.74	0.84	0.52
	I have a warm, satisfying, and trusting relationship with others.	0.73		
	I have a feeling of continued development.	0.80		
	I am concerned about the welfare of others.	0.81		
	I have a positive attitude toward myself.	0.79		
Organizational Citizenship behavior	I keep abreast of changes in the organization.	0.70	0.81	0.51
	I attend functions that are not required but that helps the company image.	0.79		
	I attend and participate in meetings regarding the organization.	0.79		
	I keep up with developments in the company.	0.75		

**Table 3 ijerph-17-08856-t003:** Descriptive Statistics and Correlations of Variables.

Variables	1	2	3	4
1. Perceived physical ability	1			
2. physical self-presentation confidence	0.14 **	1		
3. Psychological well-being	0.26 ***	0.24 ***	1	
4. Organizational citizenship behavior	0.08	0.33 ***	0.30 ***	1
Mean	2.98	3.09	3.33	3.29
Standard deviation	0.96	1.00	0.99	0.95

Note. All items were a 5-point Likert-type scale. ** *p* < 0.01. *** *p* < 0.001, two-tailed.

**Table 4 ijerph-17-08856-t004:** Path Coefficients between Perceived Physical Ability, Physical Self-presentation Confidence, Psychological Wellbeing, and Organizational Citizenship Behavior.

Hypothesis	Path	*β*	*SE*	*P*
H1	Perceived physical ability → Psychological well-being	0.24	0.05	0.001
H2	Physical self-presentation confidence → Psychological well-being	0.03	0.05	0.593
H3	Perceived physical ability → Organizational citizenship behavior	0.23	0.05	0.001
H4	Physical self-presentation confidence → Organizational citizenship behavior	0.31	0.05	0.001
H5	Psychological well-being → Organizational citizenship behavior	0.27	0.06	0.001

χ^2^ = 194.01, *df* = 129, CFI = 0.97, TLI = 0.97, RMSEA = 0.03, RMR = 0.04.

**Table 5 ijerph-17-08856-t005:** The Results of moderating effects of leisure-time physical activity.

Path	Δχ^2^	Path Coefficients			
		Regular Exercise Group		Non-Regular Exercise Group	
		*β*	*SE*	*β*	*SE*
PPA → PWB	5.59 *	0.46 ***	0.10	0.12	0.06
PSC → PWB	1.56	0.40 ***	0.08	0.23 *	0.07
PPA → OCB	2.16	−0.11	0.10	0.06	0.05
PSC → OCB	9.66 ***	0.44 ***	0.09	0.09	0.06

Note. PPA = perceived physical ability, PSC = physical self-presentation confidence, PWB = psychological well-being, OCB = organizational citizenship behavior, * *p* < 0.05, *** *p* < 0.001.

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
