# Peer review of "Structural Relationship among Physical Self-Efficacy, Psychological Well-Being, and Organizational Citizenship Behavior among Hotel Employees: Moderating Effects of Leisure-Time Physical Activity"

_ijerph, 2020, doi:10.3390/ijerph17238856_

Round 1
Reviewer 1 Report
Manuscript No.: ijerph-993333 Manuscript Title: Physical Self-Efficacy, Psychological Well-being, and Organizational Citizenship Behavior in Hotel Employees: Moderating Effects of Leisure-Time Physical Activity First of all, the theoretical contribution of this manuscript is not enough. The impact of physical self-efficacy on the psychological well-being and organizational citizenship behavior has been well examined. From my perspective, the impacts and mechanisms of physical self-efficacy on work outcomes have been sufficiently verified. Although examining the moderating effects of leisure-time physical activity on the relationships above seems necessary among hotel employees, I do not think it is very innovative from a theoretical perspective. Thus, I highly recommend authors could highlight the importance of doing this research in the context of hospitality field. For example, comparing with other disciplines, what are the specific characters of hospitality employees? Surface acting, deep acting, or even emotional labor are much more obvious for hotel employees? The conclusion section seems not well organized. I cannot see a very clear implications for the industry. Thank you.Author Response
Response to Reviewer 1
TITLE: Structural Relationship among Physical Self-Efficacy, Psychological Well-being, and Organizational Citizenship Behavior among Hotel Employees: Moderating Effects of Leisure-Time Physical Activity
REVIEWER'S COMMENTS:
- First of all, the theoretical contribution of this manuscript is not enough.
Response: Thank you for pointing out the issue. To address your concern, we have reworked the Introduction and Discussion of our manuscript. Please see line number from 88 to 104 and 338 to 341 on page 2, 3, and 10, respectively. We hope this modification adequately addressed your concern. Thank you.
- The impact of physical self-efficacy on the psychological well-being and organizational citizenship behavior has been well examined. From my perspective, the impacts and mechanisms of physical self-efficacy on work outcomes have been sufficiently verified.
Response: Thank you for your comments!
- Although examining the moderating effects of leisure-time physical activity on the relationships above seems necessary among hotel employees, I do not think it is very innovative from a theoretical perspective. Thus, I highly recommend authors could highlight the importance of doing this research in the context of hospitality field. For example, comparing with other disciplines, what are the specific characters of hospitality employees?
Response: Per your suggestion, we have reworked the following sentences in the Introduction of our manuscript. Please see line numbers from 88 to 104 on page 2 and 3. We hope these additions adequately address your comments. Thank you.
"In sum, the current study aimed to investigate the effects of physical self-efficacy on the psychological well-being and organizational citizenship behavior among hotel employees and the moderating effects of leisure-time physical activity on the relationships between physical self-efficacy, psychological well-being, and organizational citizenship behavior. It is important to note that we wanted to differentiate the present study from the extant research by exploring the moderating roles of leisure-time physical activity in the relationships between the aforementioned variables. In fact, hotel employees are faced with a heavy workload and low wages and are commonly required to do excessive emotional labor, which affects turnover, while enduring consumers' mistreatment [13, 14]. Consequently, the present research started by acknowledging the limitation of previous studies on hotel employees' organizational citizenship behavior, neglecting the leisure-time physical activity element. We attempted to shed light on how hotel employees' perceived physical self-efficacy affects psychological well-being and organizational citizenship behavior. We also examined if the effects of perceived physical self-efficacy on physical well-being and organizational citizenship behavior differ based on employees' leisure-time physical activity levels. The current study's findings are expected to provide hotel organizations with effective measures to manage human resources by considering leisure-time physical activity programs as well as the psychological well-being of employees."
- Surface acting, deep acting, or even emotional labor are much more obvious for hotel employees? The conclusion section seems not well organized. I cannot see a very clear implications for the industry.
Response: Per your suggestion, we have added the sentences as below in our manuscript's Conclusions section. Please see line numbers from 393 to 400 and 417 to 421 on page 11. We hope these modifications adequately address your comments. Thank you.
"Hotel employees provide customers with various services such as reception, room information, luggage transportation, room reservation, room arrangement, laundry supply, and food provision. Hotel employees are faced with many challenges while performing their jobs. Facing a heavy workload, frequent changes in circumstances, lack of performance feedback, and low wages, they get easily irritated and exhausted, affecting their behavior. They may lead to turnover [14]. Moreover, hotel workers have a high intensity of emotional labor in daily-based customer service interactions regardless of their inner psychological state to comply with organizationally mandated emotional display rules [13]."
"Findings of this study also suggest that physical self-efficacy is a crucial antecedent of well-being and organizational citizenship, which leads to high employee productivity in the hotel industry. This indicates that hotel managers should pay attention to this personal trait in the recruitment process. Thus, hotel organizations are required to establish a reliable and valid physical self-efficacy test and apply it to employee recruitment as well as counseling program adjustment."
Reviewer 2 Report
ABSTRACT
The abstract should describe the content and scope of the paper and identifies the paper’s objective or hypothesis, its methodology and its findings, conclusions, or intended results. The purpose of the abstract is to report the original contributions of your research.
The abstract should begin with a brief but precise statement of the problem or issue, followed by a description of the research method and design, the major findings, and the conclusions reached. Some of these issues are not included. Information on the methodological tools used and the number of cases analysed is not included, nor is the location of the study indicated.
INTRODUCTION
Explain better from the beginning what the concept of " employees' voluntary participation" or “voluntary efforts” implies.
Explain how the paper is organized. Explain better the originality of this study. It is recommended to explain clearly how problem is going to be answered.
LITERATURE REVIEW
In general, the review of scientific literature is scarce. The theoretical framework is barely one page. It is therefore important to invest considerable time and effort into a careful literature review. I strongly recommended to improve and enrich the bibliography.
A well-constructed literature review identifies major themes associated with the topic, and demonstrate where there is agreement and disagreement about the topic. This section should identify limitations of prior research and exposes gaps in our understanding about the topic, which indicates possible directions of future inquiry of the topic. A well-constructed literature review should situate the proposed research in the context of extant literature, and it should be clearly identify ho the proposed research will create new knowledge that enhances the existing knowledge about this topic.
It is clear that the section "Hypotheses Development" exposes in detail the different hypotheses of the study and justifies them based on previous studies. Even considering the literature review included in this section, it is recommended to deepen and improve the "literature review" section.
MATERIALS AND METHODS
One critical aspect of publishing research is describing the methods used in enough detail that the experiments can be reproduced by others. The authors have described statistical tests in detail. Descriptive Statistics and Correlations of Variables are very well defined. However, some doubts arise about the methodology used
On one hand, the items that define or are included in the variables sometimes seem to be similar to each other (for example in the variable "Perceived physical ability"). It is considered appropriate to explain in more detail the well-founded reasons for including these items and not others. Other questions: who validated the survey? was it reviewed by experts? or what scientific reasons justify the inclusion of these and not other items to define the variables analyzed?
On the other hand, the authors say that “The survey questionnaire consisted of 24 items designed to measure physical self-efficacy, psychological well-being, organizational citizenship behavior, and leisure-time physical activity (see Table 2)”. However, the items included in table 2 are 18. Explain please.
Please, justify the validity of the sample size
DISCUSSION
In general this section is adequate and complete. Could be interesting to save evaluations for different methods for the Discussion section of your paper: What methodology was used in similar studies?.
Review possible recent studies on the subject after the start of the pandemic.
CONCLUSION
In general, the conclusions included are interesting and well justified. It may be interesting to discuss or include as possible future research the possible differences found between the staff of the different departments as well as in relation to the category of the hotels.
Author Response
Response to Reviewer 2
'REVIEWER'S COMMENTS:
- The abstract should begin with a brief but precise statement of the problem or issue, followed by a description of the research method and design, the major findings, and the conclusions reached. Some of these issues are not included. Information on the methodological tools used and the number of cases analysed is not included, nor is the location of the study indicated.
Response: Thank you for pointing out the issue. We have added the following sentences in the Abstract of our manuscript. Please see line numbers from 18 to 25 on page 1. We hope this addition adequately addresses your comments. Thank you.
"To achieve the research purpose, 346 hotel employees working at the room, food, and beverages and kitchen departments of 10 hotels located in Seoul, South Korea, participated in the study. The researchers visited their department meetings and provided a brief description of the present study and informed consent forms to participate in the study. After obtaining written informed consent forms, the researchers distributed the surveys and asked participants to complete them. Several statistical analyses, including descriptive analysis, a confirmatory factor analysis for examining hypothesized model's psychometric properties, and structural equation modeling for testing the hypotheses by using SPSS Ver. 23.0 and AMOS 23.0."
- Explain better from the beginning what the concept of " employees' voluntary participation" or ""voluntary efforts"" implies.
Response: Thank you for pointing out the issue. Strictly following your suggestions, we have modified the first sentence of the Introduction of our manuscript as below. Please see line numbers from 37 to 39 on page 1. We hope this addition adequately addresses your comments. Thank you.
"Due to the uncertainty of market environments, service firms are more dependent on their employees' voluntary participation, which can be 'employees' voluntary attitude and behaviors that help other coworkers, and commitment than ever before."
- Explain how the paper is organized. Explain better the originality of this study. It is recommended to explain clearly how problem is going to be answered.
Response: Per your suggestion, we have reworked the following sentences in the Introduction of our manuscript. Please see line numbers from 88 to 104 on page 3 and 4. We hope these additions adequately address your comments. Thank you.
"In sum, the current study aimed to investigate the effects of physical self-efficacy on the psychological well-being and organizational citizenship behavior among hotel employees and the moderating effects of leisure-time physical activity on the relationships between physical self-efficacy, psychological well-being, and organizational citizenship behavior. It is important to note that we wanted to differentiate the present study from the extant research by exploring the moderating roles of leisure-time physical activity in the relationships between the aforementioned variables. In fact, hotel employees are faced with a heavy workload and low wages and are commonly required to do excessive emotional labor, which affects turnover, while enduring consumers' mistreatment. Consequently, the present research started from the acknowledgment of the limitation of previous studies on hotel 'employees' organizational citizenship behavior neglecting the leisure-time physical activity element. We attempted to shed light on how hotel 'employees' perceived physical self-efficacy affects psychological well-being and organizational citizenship behavior. We also examined if the effects of perceived physical self-efficacy on physical well-being and organizational citizenship behavior differ based on 'employees' leisure-time physical activity levels. The current study's findings are expected to provide hospitality firms with effective measures to manage human resources by considering leisure-time physical activity programs as well as the psychological well-being of employees."
- In general, the review of scientific literature is scarce. The theoretical framework is barely one page. It is therefore important to invest considerable time and effort into a careful literature review. I strongly recommended to improve and enrich the bibliography.
Response: Thank you for pointing out the issue. Strictly following your suggestions, we reworked the Literature Review and Hypotheses Development of our manuscript. Please see line number from 105 to 199 on pages from 3 to 5. We hope this modification adequately addresses your comments. Thank you.
- A well-constructed literature review identifies major themes associated with the topic, and demonstrate where there is agreement and disagreement about the topic. This section should identify limitations of prior research and exposes gaps in our understanding about the topic, which indicates possible directions of future inquiry of the topic. A well-constructed literature review should situate the proposed research in the context of extant literature, and it should be clearly identify ho the proposed research will create new knowledge that enhances the existing knowledge about this topic.
Response: Per your suggestion, we have modified the Literature Review and Hypotheses Development of our manuscript. Please see line number from 105 to 199 on pages from 3 to 5. We hope this modification adequately addresses your comments. Thank you.
- It is clear that the section "Hypotheses Development" exposes in detail the different hypotheses of the study and justifies them based on previous studies. Even considering the literature review included in this section, it is recommended to deepen and improve the "literature review" section.
Response: Per your suggestion, we have reworked the Literature Review and Hypotheses Development of our manuscript. Please see line number from 105 to 199 on pages from 3 to 5. We hope this modification adequately addresses your comments. Thank you.
- One critical aspect of publishing research is describing the methods used in enough detail that the experiments can be reproduced by others. The authors have described statistical tests in detail. Descriptive Statistics and Correlations of Variables are very well defined. However, some doubts arise about the methodology used. On one hand, the items that define or are included in the variables sometimes seem to be similar to each other (for example in the variable "Perceived physical ability"). It is considered appropriate to explain in more detail the well-founded reasons for including these items and not others. Other questions: who validated the survey? was it reviewed by experts? or what scientific reasons justify the inclusion of these and not other items to define the variables analyzed?
Response: Per your suggestion, the items used in the current study were adopted from the prior studies that were well validated [12, 20, 28]. Moreover, an expert group offour professors whose expertise lies in hospitality management, sport management, and measurement reviewed the items’ relevance, representativeness, and clarity. Accordingly, we have added the following sentence in the Methods of our manuscript. Please see line numbers from 244 to 246 on page 6. We hope this addition adequately addresses your comments. Thank you.
"The items translated from English into Korean through back-translation process were reviewed by an expert group, including four professors whose expertise lies in hospitality management, sport management, and measurement reviewed the items’ relevance, representativeness, and clarity."
- On the other hand, the authors say that "The survey questionnaire consisted of 24 items designed to measure physical self-efficacy, psychological well-being, organizational citizenship behavior, and leisure-time physical activity (see Table 2)". However, the items included in table 2 are 18. Explain please.
Response: The initial survey questionnaire contained 24 items, including five demographic and leisure-time physical activity items, 18 items measuring physical self-efficacy, psychological well-being, and organizational citizenship behavior. However, one item measuring perceived physical ability was removed due to low factor loading (< .70). As a result, a total of 23 items were used for the subsequent data analysis in the current study. Please refer to the line number from 262 to 273 on page 7 of our manuscript. Thank you.
- Please, justify the validity of the sample size
Response: Thank you for pointing out the issue. We have reviewed previous studies on organizational citizenship behavior in the context of hospitality management. Studies on a similar topic (e.g., Aguiar-Quintana, Araujo-Cabrera, & Park, 2020; Zhao & Zhou, 2020) among hotel employees have used a sample size between 200 and 300. Thus, our sample size of 346 was considered to be adequate. We hope the above justification was appropriate. Thank you.
- In general this section is adequate and complete. Could be interesting to save evaluations for different methods for the Discussion section of your paper: What methodology was used in similar studies?
Response: Thank you for your comments. Studies examining the similar issue employed SEM as the main analytical method to test their hypotheses.
- Review possible recent studies on the subject after the start of the pandemic.
Response: Thank you for pointing out the issue. Since the pandemic situation is beyond the scope of our research purpose, we have added the following sentences in the Conclusions of our manuscript as a study limitation. Please see line number from 432 to 435 on page 11. We hope this addition adequately addresses your comments. Thank you.
"Lastly, because the data analyzed in the current study was collected in the pre-Covid era (i.e., summer of 2018), interpretation of the results requires caution as hotel employee’s perception toward the leisure-time activity may not be the same as that is during the Covid era."
- In general, the conclusions included are interesting and well justified. It may be interesting to discuss or include as possible future research the possible differences found between the staff of the different departments as well as in relation to the category of the hotels.
Response: Thank you for pointing out the issue. Strictly following your suggestion, we have added the following sentence in the Conclusions of our manuscript. Please see line number from 429 to 432 on page 11. We hope this addition adequately addresses your comments. Thank you.
"Third, since the perception toward work environment and workload for hotel employees may be different, it would be interesting to see if the effects of perceived self-efficacy on psychological well-being and organizational citizenship behaviors differ by hotel employees’ work responsibilites."
Reviewer 3 Report
Introduction should take into account the importance of the researched phenomenon in light of the COVID-19 hospitality context, and whether it will become even more pronounced or not, as well as state to what extent has the context changed from the point of data research collection to the present COVID-19 service context.
Figure 1.: could you please better position the 4 dotted-line arrows? The way they have been presented to the reviewers is not proper.
Line 378: please reformulate the beginning. It should begin with the most important argument for the article, e.g. that employees work is performed in closed spaces and repetitive in nature. Also the next sentence needs to be clearly connected to the first sentence.
Line 368: employees play an essential role in achieving customer satisfaction and loyalty, but this link is not direct- they do so by creating a memorable experience for customers. See for example: Keiningham et al. (2019) and Dressler and Paunovic (2019)
Line: 380: please delete and reformulate “our findings” to “findings of this study”, or similar.
Author Response
Response to Reviewer 3
REVIEWER’S COMMENTS:
- Introduction should take into account the importance of the researched phenomenon in light of the COVID-19 hospitality context, and whether it will become even more pronounced or not, as well as state to what extent has the context changed from the point of data research collection to the present COVID-19 service context.
Response: Thank you for pointing out the issue. Since a pandemic situation like the COVID-19 is beyond the research scope of the present study, we have stated the issue as a research limitation in the Conclusions of our manuscript as follows. Please see line numbers from 432 to 435 on page 11. We hope this addition adequately addresses your comments. Thank you.
“Lastly, because the data analyzed in the current study was collected in the pre-Covid era (i.e., summer of 2018), interpretation of the results requires caution as hotel employee’s perception toward the leisure-time activity may not be the same as that is during the Covid era..”
- Figure 1: could you please better position the 4 dotted-line arrows? The way they have been presented to the reviewers is not proper.
Response: Thank you for pointing out the issue. We have better positioned the research model in the Literature Review and Hypotheses Development section. Please see line numbers from 200 to 201 on page 5. Thank you.
- Line 378: please reformulate the beginning. It should begin with the most important argument for the article, e.g. that employees work is performed in closed spaces and repetitive in nature. Also the next sentence needs to be clearly connected to the first sentence.
Response: Per your suggestion, we have added the sentences as below in the Conclusions of our manuscript. Please see line numbers from 393 to 400 on page 11. We hope these modifications adequately address your comments. Thank you.
“Hotel employees provide customers with various services such as reception, room information, luggage transportation, room reservation, room arrangement, laundry supply, and food provision. Hotel employees are faced with many challenges while performing their jobs. Facing a heavy workload, frequent changes in circumstances, lack of performance feedback, and low wages, they get easily irritated and exhausted, affecting their behavior. They may lead to turnover [14]. Moreover, hotel workers have a high intensity of emotional labor in daily-based customer service interactions regardless of their inner psychological state to comply with organizationally mandated emotional display rules [13].”
- Line 368: employees play an essential role in achieving customer satisfaction and loyalty, but this link is not direct- they do so by creating a memorable experience for customers. See for example: Keiningham et al. (2019) and Dressler and Paunovic (2019)
Response: Thank you for pointing out the issue. Strictly following your suggestions, we modified the sentence as below in the Discussion of our manuscript as follows. Please see line numbers from 381 to 383 on page 10 in the revised manuscript. We hope this modification adequately addresses your comments. Thank you.
“Hospitality services inherently require frequent and intimate interactions with customers, and employees play an essential role in achieving customer satisfaction and loyalty by creating a memorable experience for customers [42].”
- Line: 380: please delete and reformulate “our findings” to “findings of this study”, or similar.
Response: Per your suggestion, we have changed “Our findings” to “Findings of this study” in the Conclusions of our manuscript. Please see line numbers from 385 to 386 on page 11. We hope this modification adequately addresses your comments. Thank you.
Reviewer 4 Report
The paper is an outcome of well-designed work and gives interesting comparisons of two group of hotel employees with different level of level of leisure-time physical activity.
However, the following considerations would help enriching the paper further:
1. The authors should furnish recommendations for hotel companies and further elaborate on theoretical implications. Managerial implications are short and are in need of explanation and rationalization.
2. Please reconsider your research title. The title should be specifically addressing the topic of the paper and the relationship between all these variables.
3. I would suggest the authors to provide a figure to better and clearly reveal path coefficients in Table 4 and the results of moderating effects of leisure-time physical activity.
4. Why did you select mere physical self-efficacy as the antecedent of psychological well-being? Please provide theoretically-based argument for your study.
5. Given a few not strict-logic definitions, it would be nice that they can be reunified, e.g. sense of well-being in line 337.
6. Reorganize the contents in Abstract as it did not highlight your topic, conceptual model and gap fillings.
I feel there is potential here if all these concerns can be addressed. Good luck on your research!
Author Response
Response to Reviewer 4
REVIEWER'S COMMENTS:
1 The paper is an outcome of well-designed work and gives interesting comparisons of two groups of hotel employees with different levels of leisure-time physical activity. However, the following considerations would help to enrich the paper further:
Response: Thank you for recognizing the merits of our manuscript. Following your suggestions, we have modified, added, and reworked the sentences in our manuscript. Thank you.
- The authors should furnish recommendations for hotel companies and further elaborate on theoretical implications. Managerial implications are short and are in need of explanation and rationalization.
Response: Per your suggestion, we have added the following sentences in the Conclusions of our manuscript. Please see line numbers from 418 to 422 and from 430 to 436 on pages 11 and 12, respectively. We hope these additions adequately address your suggestions. Thank you.
"Findings of this study also suggest that physical self-efficacy is a crucial antecedent of well-being and organizational citizenship, which leads to high employee productivity in the hotel industry. This indicates that hotel managers should pay attention to this personal trait in the recruitment process. Thus, hotel organizations are required to establish a reliable and valid physical self-efficacy test and apply it to employee recruitment as well as counseling program adjustment."
"Third, since the perception toward work environment and workload for hotel employees may be different, it would be interesting to see if the effects of perceived self-efficacy on psychological well-being and organizational citizenship behaviors differ by hotel employees' work responsibilities. Lastly, because the data analyzed in the current study was collected in the pre-Covid era (i.e., summer of 2018), interpretation of the results requires caution as hotel employee's perception toward the leisure-time activity may not be the same as that is during the Covid era."
- Please reconsider your research title. The title should be specifically addressing the topic of the paper and the relationship between all these variables.
Response: Thank you for your suggestion. Strictly following your suggestion, we have changed the title of our manuscript to "Structural Relationship among physical self-efficacy, psychological well-being, and organizational citizenship behavior among hotel employees: Moderating Effects of Leisure-Time Physical Activity" on the first page of our manuscript. Please refer to the research title on page 1. We hope this modification adequately addresses your suggestion. Thank you.
- I would suggest the authors provide a figure to better and clearly reveal path coefficients in Table 4 and the results of moderating effects of leisure-time physical activity.
Response: Per your suggestion, we have added Fig 2, which shows the path coefficients of the research hypotheses in our manuscript's Results section. Please see line numbers from 288 to 291 on page 8. We hope this addition adequately addresses your comments. Thank you.
- Why did you select mere physical self-efficacy as the antecedent of psychological well-being? Please provide theoretically-based argument for your study.
Response: Thank you for your question. In response to your question, we theorized physical self-efficacy as the driver because this construct is a form of self-efficacy and refers to an individual's perceived level of competence related to physical tasks. Physical self-efficacy, which consists of perceived physical ability and physical self-presentation confidence, has been reported to reduce negative emotional states such as anxiety and depression and improve self-concept, self-esteem, and cognitive ability. More importantly, physical self-efficacy affects what activity a person chooses, how much effort the person should make, and how much persistence the person should have in the face of difficulties, determining the present and future behaviors. Given this, numerous prior studies continuously reported that self-efficacy positively impacts well-being and organizational citizenship behavior among hospitality workers. Thus, we selected physical self-efficacy as the antecedent of psychological well-being in our study. We hope this adequately addresses your comment. Thank you.
- Given a few not strict-logic definitions, it would be nice that they can be reunified, e.g. sense of well-being in line 337.
Response: Per your suggestion, we have modified the following sentence in the Discussion of our manuscript. Please see line numbers from 350 to 353 on page 10. We hope this modification adequately addresses your comment. Thank you.
"Thus, it might be necessary for hospitality firms to provide their workers with a working environment that can enhance perceived physical ability, which leads to an increased sense of psychological well-being while minimizing an organizational atmosphere that can adversely affect the employees."
- Reorganize the contents in Abstract as it did not highlight your topic, conceptual model and gap fillings
Response: Per your suggestion, we have modified the Abstract of our manuscript. Please see line numbers from 13 to 32 on page 1. We hope this modification adequately addresses your comments. Thank you.
- I feel there is potential here if all these concerns can be addressed. Good luck on your research!
Response: Thank you for your valuable comments, which definitely helped us improve our manuscript quality!
Reviewer 5 Report
It is a very necessary study for service workers. Overall, the flow of the paper is fine, but please check and review a few things below.
1. I would like to clarify more about the differences other than the moderating roles of leisure-time physical activity in the introduction.
2. In the research model, I would like to show the arrows and dotted lines related to H6 more clearly. It is difficult to understand because the text and pictures are separated.
3. It would be better to think more about the limitations of the study. There seems to be a difference depending on the work department. In particular, the stress level and stability of kitchen and room workers will be different. Also, there may be differences according to position. More detailed research in the hotel seems to be needed.
4. The reader's understanding will be better if the result of the analysis of the basic model of the entire data is expressed in a picture that can be seen at a glance.
5. It seems that it is necessary to organize the parts where the columns are too wide in the reference arrangement.
6. Excluding studies that are too old are not in principle, it is recommended to update to the latest literature.
Author Response
Response to Reviewer 5
REVIEWER’S COMMENTS:
- It is a very necessary study for service workers. Overall, the flow of the paper is fine, but please check and review a few things below.
Response: Thank you for recognizing the merits of our manuscript. Strictly following your suggestions, we have modified, added, and reworked the sentences in our manuscript. Thank you.
- I would like to clarify more about the differences other than the moderating roles of leisure-time physical activity in the introduction.
Response: Per your suggestion, we have added the following sentences in the Introduction of our manuscript. Please see line numbers from 77 to 80 on page 2. We hope this addition adequately addresses your comment. Thank you.
“Additionally, Kekäläinen, Freund, Sipilä, and Kokko [12] reported that walking was positively related to psychological and social well-being. They also found that endurance training was associated with subjective health and well-being. The results clearly suggest that , suggesting that leisure activities were associated with psychological well-being.”
- In the research model, I would like to show the arrows and dotted lines related to H6 more clearly. It is difficult to understand because the text and pictures are separated.
Response: Per your suggestion, we have re-located the research model in the Literature Review and Hypotheses Development of our manuscript. Please see line numbers from 200 to 201 on page 5 for the hypothesized model. We hope this modification adequately addresses your comment. Thank you.
- It would be better to think more about the limitations of the study. There seems to be a difference depending on the work department. In particular, the stress level and stability of kitchen and room workers will be different. Also, there may be differences according to position. More detailed research in the hotel seems to be needed.
Response: Per your suggestion, we have added the following sentences in the Conclusions of our manuscript. Please see line numbers from 394 to 401 and from 433 to 436 on page 11 and 12, respectively. We hope these additions adequately address your comments. Thank you.
“Hotel employees provide customers with various services such as reception, room information, luggage transportation, room reservation, room arrangement, laundry supply, and food provision. Hotel employees are faced with many challenges while performing their jobs. Facing a heavy workload, frequent changes in circumstances, lack of performance feedback, and low wages, they get easily irritated and exhausted, affecting their behavior. They may lead to turnover [14]. Moreover, hotel workers have a high intensity of emotional labor in daily-based customer service interactions regardless of their inner psychological state to comply with organizationally mandated emotional display rules [13].”
“Lastly, because the data analyzed in the current study was collected in the pre-Covid era (i.e., summer of 2018), interpretation of the results requires caution as hotel employee’s perception toward the leisure-time activity may not be the same as that is during the Covid era..”
- The reader's understanding will be better if the result of the analysis of the basic model of the entire data is expressed in a picture that can be seen at a glance.
Response: Per your suggestion, we have added Fig 2, which shows the path coefficients of the research hypotheses in our manuscript's Results section. Please see line numbers from 289 to 291 on page 8. We hope this addition adequately addresses your comments. Thank you.
- It seems that it is necessary to organize the parts where the columns are too wide in the reference arrangement.
Response: Per your suggestion, we have used the MS Word journal template provided by the this journal and have strictly followed its editing regulations. Please see line numbers from 443 to 554 on the page from 12 to 14. We hope this adequately addresses your comment. Thank you.
- Excluding studies that are too old are not in principle, it is recommended to update to the latest literature.
Response: Per your suggestion, we have replaced several latest articles with old references appeared in the initial manuscript. Please see numbers from 443 to 554 on the pages from 12 to 14. We hope these modifications adequately address your comments. Thank you.
Round 2
Reviewer 1 Report
Manuscript No.: ijerph-993333
Manuscript Title: Physical Self-Efficacy, Psychological Well-being, and Organizational Citizenship Behavior in Hotel Employees: Moderating Effects of Leisure-Time Physical Activity
Recommendation: Minor Revision
Thank you for your effort in revising this manuscript. The revised manuscript reads well. However, there is still room for improvement. From page 3 to page 5, more theoretical foundation is required for hypotheses development. You mentioned social cognitive theory. Yet, you don’t have any theoretical background about the theory and the hypotheses. The rationale behind the hypotheses is not very clear and therefore, not entirely convincing. How the hypotheses were developed are not mentioned. I do not grab the hypotheses and the model of the paper. They are all missing. In addition, most literatures cited are quite outdated. For instance, Williams (1991).
Author Response
Response to Reviewer 1
REVIEWER'S COMMENTS:
- Thank you for your effort in revising this manuscript. The revised manuscript reads well. However, there is still room for improvement. From page 3 to page 5, more theoretical foundation is required for hypotheses development. You mentioned social cognitive theory. Yet, you don’t have any theoretical background about the theory and the hypotheses. The rationale behind the hypotheses is not very clear and therefore, not entirely convincing.
Response: Thank you for pointing out the issue. To address your concern, we have added social cognitive theory and theoretical background about the theory in the Literature Review and Hypotheses Development of our manuscript. Please see line numbers from 107 to 111 on page 3. We hope this modification adequately addressed your concern. Thank you.
- How the hypotheses were developed are not mentioned. I do not grab the hypotheses and the model of the paper. They are all missing.
Response: Per your suggestion, we have slightly modified the sentence in the Literature Review and Hypotheses Development of our manuscript. Please see line numbers from 152 to 154 on page 4. We hope this modification adequately addresses your comments. Thank you.
- In addition, most literatures cited are quite outdated. For instance, Williams (1991).
Response: For Williams and Anderson [24], this article was cited as we wanted to introduce the theoretical background of the subdivision of organizational citizenship behavior (i.e., OCB-I, OCB-O). Thus, it is necessary to include the article in our manuscript despite being a little bit outdated. We hope this adequately addresses your comments. Thank you.
Reviewer 2 Report
Thank you for the effort in reviewing and improving your article. All questions raised have been satisfactorily resolved.
However, I believe that a small improvement is still necessary. I believe that a brief review of the introduction would be necessary prior to publication. I recommend including bibliographic review regarding social cognitive theory. it is also recommended to improve the justification of the hypothesis.
Author Response
Response to Reviewer 2
REVIEWER'S COMMENTS:
- Thank you for the effort in reviewing and improving your article. All questions raised have been satisfactorily resolved.
Response: Thank you for your valuable comments!
- However, I believe that a small improvement is still necessary. I believe that a brief review of the introduction would be necessary prior to publication. I recommend including a bibliographic review regarding social cognitive theory.
Response: Per your suggestion, we have added the definition of social cognitive theory. Please see line numbers from 107 to 111 on page 3. We hope this addition adequately addresses your comments. Thank you.
- it is also recommended to improve the justification of the hypothesis.
Response: Per your suggestion, we have revised the sentences in the Literature Review and Hypotheses Development of our manuscript. Please see line numbers from 152 to 154 on page 4. We hope this modification adequately addresses your comments. Thank you.
Reviewer 4 Report
The author has worked hard and has made great progress compared to the original version. Through revision processes, this paper seems to have become more interesting and novelty. The literature part is also very full, and the organization is relatively clear.
Through the reading of this article, we can better to investigate the effects of physical self-efficacy on the psychological well-being and organizational citizenship behavior among hotel employees and the moderating effects of leisure-time physical activity on the relationships
All in all, this paper is well prepared.
Author Response
Response to Reviewer 4
REVIEWER'S COMMENTS:
- The author has worked hard and has made great progress compared to the original version. Through revision processes, this paper seems to have become more interesting and novelty. The literature part is also very full, and the organization is relatively clear. Through the reading of this article, we can better to investigate the effects of physical self-efficacy on the psychological well-being and organizational citizenship behavior among hotel employees and the moderating effects of leisure-time physical activity on the relationships. All in all, this paper is well prepared:
Response: Thank you for recognizing the merits of our manuscript!
Reviewer 5 Report
It seems to have been well modified as reviewed.
Lastly, it would be nice to finish by adding a check for overall typos. The researchers worked hard to write and edit the paper. It seems to be a good paper that can be helpful academically and practically.
Author Response
Response to Reviewer 5
REVIEWER’S COMMENTS:
- It seems to have been well modified as reviewed. Lastly, it would be nice to finish by adding a check for overall typos. The researchers worked hard to write and edit the paper. It seems to be a good paper that can be helpful academically and practically:
Response: Thank you for recognizing the merits of our manuscript! Per your suggestion, we have checked for overall typos in our manuscript. Thank you.